# A Grid-Based Approach for Measuring Similarities of Taxi Trajectories

**DOI:** 10.3390/s20113118

**Published:** 2020-05-31

**Authors:** Wei Jiao, Hongchao Fan, Terje Midtbø

**Affiliations:** 1School of Remote Sensing and Information Engineering, Wuhan University, Wuhan 430072, China; jiaowei2017@whu.edu.cn; 2Department of Civil and Environmental Engineering, Faculty of Engineering, Norwegian University of Science and Technology, 7491 Trondheim, Norway; terjem@ntnu.no

**Keywords:** similarity measurement, trajectory data, spatial distribution, grid-based approach

## Abstract

Similarity measurement is one of the key tasks in spatial data analysis. It has a great impact on applications i.e., position prediction, mining and analysis of social behavior pattern. Existing methods mainly focus on the exact matching of polylines which result in the trajectories. However, for the applications like travel/drive behavior analysis, even for objects passing by the same route the trajectories are not the same due to the accuracy of positioning and the fact that objects may move on different lanes of the road. Further, in most cases of spatial data mining, locations and sometimes sequences of locations on trajectories are most important, while how objects move from location to location (the exact geometries of trajectories) is of less interest. For the abovementioned situations, the existing approaches cannot work anymore. In this paper, we propose a grid aware approach to convert trajectories into sequences of codes, so that shape details of trajectories are neglected while emphasizing locations where trajectories pass through. Experiments with Shanghai Float Car Data (FCD) show that the proposed method can calculate trajectories with high similarity if these pass through the same locations. In addition, the proposed methods are very efficient since the data volume is considerably reduced when trajectories are converted into grid-codes.

## 1. Introduction

With the development of sensor technology for positioning, ubiquitous Global Navigation Satellite Systems (GNSS) enabled mobile devices to generate huge amounts of trajectory data [1,2]. These trajectory data are sequences containing various information such as location, time, speed and direction, which enables the rapid development of Location-Based Services (LBSs) and applications [3,4,5,6]. Similarity measurements, as one of the crucial tasks of data mining, have been widely used in the location-based applications to find all similar trajectories from a large collection [7,8]. Through similarity measurements of trajectory data, valuable information can be mined from large collections of trajectories, such as traffic flows and hot routes [9,10]. It is very useful for many applications, such as urban hotspot detection [11,12], behavior model analysis [13,14], traffic monitoring and prediction [15,16,17], urban planning [18,19] and location optimization [20,21], etc. Advances in location-based applications are increasingly creating new, sophisticated mechanisms that can foster the exchanges of information among travels to provide better services and promote a sustainable economy.

The information concerned by the similarity calculation is inconsistent in different types of applications. For example, in many sports such as table tennis and football, it is very useful for sport researchers to analyze the movement patterns of top players. The trajectories are classified by using attributes of the trajectory (e.g., direction, speed, curvature, and other descriptors) to find the similar trajectories of objects’ (e.g., players, balls) motions [22,23]. In applications dealing with animal migration patterns and urban emergency, the calculation of similarity needs to focus not only on the spatial location information of the trajectory, but also on the temporal attribute information [24,25]. Further, for location-based spatial trajectories, several applications mainly focus on the location of the routes, while neglecting the time, direction and other information. This is illustrated by the following three different types of application scenarios.

Firstly, in the location recommendation service, the trajectory similarity of moving objects resembles the paths’ or locations’ similarity. Combing this information, the frequent locations and travel habits can be mined from the large collection of trajectories, and subsequently utilized to serve the ordinary traveler [26,27,28]. Secondly, in the process of public transportation planning, the similarity measurement can be used by the public transportation company to analyze the aggregated trajectories of residents and obtain typical travel routes (e.g., knowing which routes are accessed densely) [29]. The convenience of public transport can be improved by adjusting the public transport network or adding new routes. The goal is to make the best use of the bus services and maximize the convenience of commuters so as to better serve urban residents. Thirdly, in travel behavior analysis or groups analysis, such as taxi driving behaviors, analyzing the paths’ similarity of high-income taxis can provide guidance for ordinary taxi drivers [30,31]. The goal is to reduce the cruising time of taxis and increase revenue. In these kinds of location-based applications, the calculation of trajectories’ similarity mainly requires location information, while the speed, direction and time information are not critical. It motivates us to find a suitable method to measure the spatial similarity, which is the important part of the similarity analysis of the spatial dataset.

Methods to measure similarity between trajectories have been well studied on movement pattern related to time sequences. For example, a native similarity measure is the Euclidean distance, calculated as the sum of distances between ordered pairs of sampling points in two trajectories [32]. However, due to the different sampling rates or different speeds in trajectory data, some sampling points may not be well aligned in space. Aiming to overcoming this kind of problem, several algorithms have been proposed [33,34,35]. For example, the Closest-Pair Distance is used to find the trajectories closest to a given trajectory in a spatial network [36]. The Hausdorff distance is a measure of the similarity between two sets of points [37]. The Dynamic Time Warping (DTW) algorithm allows some sample points to be repeated to achieve optimal alignment [38]. These types of methods struggle with the different sampling rates, trajectory length and the outliers, and achieves good results in trajectory clustering, map matching and other applications. However, such methods focus too much on the details of the trajectory. Hence, small deviations can have a big impact on the results (i.e., causing a large distance between trajectory sequences). Additionally, due to the accuracy of GNSS measurements and the fact that cars may drive on different lanes on the same road, the shape and distance between sequences of the same route may be different, leading to inaccuracy in the calculation results.

Another type of similarity method is calculated based on the road network. The real distance on road network between POIs is used to measure the similarity between the trajectories [1,39,40]. Xia et al. [41] and Abraham et al. [42] mapped the trajectory data to the road network and proposed a new trajectory clustering method, which calculated the similarity of the trajectory by calculating the length of the matching road segment. In applications of spatial data mining based on trajectory data, this type of method is an improvement over previous measurements and can be used for vehicle navigation and route recommendation. However, this kind of method requires detailed road information which are lacking in the vehicle trajectory data. Due to instability in the GNSS signals, errors will inevitably occur during the road matching process (especially in areas with dense road). Subsequently, this may reduce the accuracy of the similarity calculations.

Overall, neither the method of calculating the distance between sequences nor the indirect method of expressing the trajectory in other forms can satisfy the spatial similarity measure of the location-based application in our research. Our key observation is that trajectories are considered similar if they pass through multiple identical locations/places. For example, suppose the triangle area in the Figure 1 is a square. *T1* describes an object moving on road segment *Road1* and then making a left turn to the road segment *Road2*. *T2* describes an object moving on road segment *Road1*, making a left turn to the road segment *Road3* and then moving on the *Road2* (when the current road is congested, the temporary diversion is selected). Trajectories *T1* and *T2* are similar in location-based studies because they passed through the same locations/places. However, if we consider the distance between sequences, shape or the road information of the trajectory, their similarity is rather small. In this case, it is unnecessary to know the characteristics of the trajectory in detail, although there may be some interesting issues in those trajectories. Hence, the spatial similarity measurement of trajectory data mainly focuses on the common locations or places, and these locations should be the meaningful areas (e.g., a square, commercial streets, office areas), not just points with latitude and longitude. We contend that the location-based similarity analysis is very meaningful, especially if we have some particularly interesting locations for analysis (e.g., travel preferences analysis).

Due to the facts mentioned above, the existing approaches may be inappropriate in Location-based applications. This paper proposes a grid-based approach called Spatial Grid Coding Distance (SGCD) for measuring similarity of trajectories. Instead of directly using point-list, only grid-codes based on the location of trajectories are selected for similarity measurement. The algorithm converts the trajectories by setting the appropriate grid size, which not only neglects the shape details of the trajectories, but also does not require the alignment between the sampling points. Additionally, the recent development of location-based applications shows that the demand for spatial similarity calculations is becoming more and more diverse. Instead of focusing on one particular model in the process of spatial similarity calculation, we should consider multi-scale query processing [43,44]. The characteristics of common locations and self-intersection of trajectories are considered, respectively. Consequently, two spatial similarity calculation methods are proposed in this paper, including common location similarity and structural similarity. Each kind of similarity measurement algorithm can be used separately during the knowledge discovery process.

The remainder of the paper is organized as follows. Section 2 describes the framework containing the workflow of data processing. The main principles of our approach are introduced in Section 2.1, Section 2.2 and Section 2.3. Section 3 develops an experiment in order to test our proposed algorithm. Finally, Section 4 concludes the paper and outline further works.

## 2. The Grid-Based Approach for Similarity Measurement

A trajectory is a sequence of time-stamped sampling points, and its location information can be represented as a list of geographic coordinates. In order to emphasize the impact of locations while eliminating the disturbance of geometric details of trajectories in the process of similarity measurement, the trajectory was represented with grid-codes in our algorithm. The algorithm runs through two main phases, followed by an algorithmic verification phase (Figure 2). Phase I consists of grid-based trajectory conversion. First, a rule for determining the appropriate grid size is defined in order to convert the study area into grids (Section 2.1). Second, the original trajectory data (i.e., *Traj-1*, *Traj-2*, …, *Traj-n*) is superimposed on a coded grid to achieve trajectory conversion (Section 2.2). After the trajectory conversion, the original trajectory data is represented by grid-codes and converted into *R_Traj-1*, *R_Traj-2*, …, *R_Traj-n*. In Phase II, through algorithm design, different types of trajectory similarity measurements are realized in Section 2.3. The following sections explain the workflow in detail.

### 2.1. Grid Generation

#### 2.1.1. The Determination of Grid Cell Size

One of the issues of converting the study area from vector feature to regular grid is the determination of the size of the grid cells. If the size is too large, the integrality of trajectory information can be ignored. This will lead to the lack of some basic feature information of trajectories, making the result of the similarity calculation unreliable. If the size is too small, too much attention will be paid to the details of the trajectories. This does not eliminate the effect of the details on the trajectory, leading to small deviations that can have a significant impact on the results. What is more, the computational cost will be very high. Therefore, choosing an appropriate size for the grid cell ensures that the model achieves better performance and acceptable accuracy, considering that the road network in a city is an important indicator of urban development level [45]. Developed areas normally have denser roads and more points of interest (e.g., hospitals, schools), while less developed areas have a sparse road network and fewer meaningful places. Since our research focuses on the spatial similarity calculation based on common locations/places, road network spacing in the city is used as the basis for sizing the grid cells.

However, there are two problems that should be solved in this process. Firstly, the road network in the city can be divided into several levels (e.g., main road, trunk road, secondary trunk road and branch road). Hence, the selection of the level of the road network is the first problem. In general, lower level road networks represent smaller road networks spacing. The lowest level road (i.e., branch road) network will fragment a meaningful place. The blue area in Figure 3 is Square A: the branch road network contains many small paths, which can divide Square A into many fragmented patches (*A1*, *A2*, *A3*, *A4*, *A5*). Further, in similarity analysis, trajectories are normally considered to be similar because they pass by a few identical locations/places. Consequently, the lower-level road network should be selected based on the fact that meaningful places are not fragmented. In this way, the sampling points at the same place can be prevented from being divided into different areas, thereby improving the accuracy of the calculation results. 

The second problem is to solve the imbalanced spatial distribution of the road network. Many studies have shown that urban road network density is strongly related to spatial structure [46,47]. Cai et al. used the kernel density estimation to assess the spatial pattern of road density and clearly reflected that the kernel densities of roads decreased with greater distance from urban core areas or central business districts [48]. In fact, many cities have more than one central business district. There are three main theories of urban spatial structure, namely the concentric circle model [49], sector theory [50] and multi-core theory [51]. In terms of the number of urban centers, the urban spatial structure has two types: monocentric and polycentric structure. The density of road networks in downtown areas are higher than in suburban areas in a mono-centered city. In poly-centered cities, densities of the road network in city centers are higher than those in other areas. The road network density is strongly related to the level of urban development, and areas with high road network density have more meaningful locations. Since our research is based on spatial similarity calculation for meaningful locations, the number of locations in each grid cell should be balanced. For this reason, in the process of determining the grid size, the areas with high road network density and the areas with low road network density should be treated separately.

According to the road network density, the study area can be divided into several sub-areas (such as *R_1_*, *R_2_*, …, *R_i_*). Generally, a city can be divided into suburban and downtown areas. The kernel density estimation is used to calculate the road network density in the sub-region *R_i_*. In the kernel density analysis, the bandwidth is an important factor which directly affects the results of the density analysis. This study used the calculation formula proposed by Silverman [52] (Equation (1)). A raster data of road network density is generated by kernel density calculation, and its attribute value is the road network density within the range of cells. The cell can be called a density unit in our research.
(1)$$bandwidth=0.9*\min\left(SD,\sqrt{\frac{1}{\ln(2)}}*D_{m}\right)*n^{-0.2}.$$
where *SD* is the standard deviation, *D_m_* is the median distance. This formula weighs the two parameters of standard deviation and median distance and takes the minimum of the two to contribute to the final calculation.

According to the relationship between the road network density and the road network spacing, the average value of the road network spacing in the sub-region *R_i_* is calculated as the grid size of this area. So, the grid size *G* (*road_level_, R_i_*) can be calculated according to the following equation:(2)$$G(road_{level},R_{i})=\frac{\sum_{j=0}^{m_{i}}f(road_{level},R_{i},j)}{m_{i}},$$
where the *f*(*road_level_, R_i_, j*) is the road network spacing in the *j-th* density unit of raster data (kernel density analysis result) in sub-area *R_i_*; the *road_level_* is the selected level of road network; *m_i_* is the number of density unit in subarea *R_i_*. According to Miyagawa [53], *f*(*road_level_, R_i_, j*) can be calculated according to the following equation:(3)$$f(road_{level},R_{i},j)=\frac{2}{density\_road(road_{level},R_{i},j)}.$$
where the *density_road* (*road_level_, R_i_, j*) is the density of the selected road network in the *j*-*th* density unit of sub-area *R_i_*.

#### 2.1.2. The Generation of Grid

For an irregular convex polygonal study area, in order to divide it into a regular grid, a Minimum Bounding Rectangle (MBR) needs to be established. The MBR is a rectangle oriented to the x and y axes and it is one of the most frequently used methods to express the geographic feature or a geographic dataset. The MBR is determined by two coordinates: *Xmin*, *Ymin* and *Xmax*, *Ymax*, which are obtained based on the geographic coordinate range of the study area. Assuming there exists a study area, the length of the MBR is x and the width is y. According to the method proposed in the previous section, the size of the grid cell of the study area is determined as *δ*. The conversion of the grid is as follows:(4)$$\mu_{x}=\mu_{y}=\delta ,$$
(5)$$M=\frac{x}{\mu_{x}},$$
(6)$$N=\frac{y}{\mu_{y}}.$$
where the $\mu_{x}$, $\mu_{y}$ are the length and width of each grid cell respectively; *M* and *N* are the numbers of the rows and columns, respectively.

According to the previous section, the study area should be divided into several sub-areas to solve the problem of spatial heterogeneity. Subsequently, separate grids are generated for each sub area before they are all merged into one grid.

### 2.2. Converting Trajectory with Grid Code

Let *T* be a trajectory in space, represented as *T* = (*p1*, *p2*, …, *pn*), where n is the number of sample points in *T* and *pi* (1 < *i* < *n*) is a sample point with geographic coordinates. These sample points are arranged in the order of sample time.

Since this study focuses on the spatial similarity of trajectory data, we only need to pay attention to the location information of the trajectory during the trajectory conversion process. As shown in the Figure 4, the trajectory data is superimposed on the coded grid of the study area. If the sample point is within the grid cell, the location information of the sample point is replaced with the grid code. By converting each sample point, the trajectory is finally converted into grid-codes.

It is worth mentioning that, for the vehicle trajectory data, the sampling frequency is 10–60 s, which may result in multiple consecutive sampling points falling in the same grid cell. To reduce data redundancy, duplicate data needs to be filtered (i.e., *num_i+1_* ≠ *num_i_*). Through the filtering of the trajectory it is possible to both reduce the redundancy and maintain the integrity of the data. This is beneficial for the data storage. The converted trajectory is shown in Figure 4. The attribute value of grid cell is the number of times that the trajectory passes through the grid cell. For grid (3,4), the attribute of the grid is five, which means that the trajectory passes through this grid cell five times. After the trajectory conversion, the location information of trajectory is expressed by the grid-codes and the spatial distribution of the trajectory is represented by the trajectory matrix.

### 2.3. The Grid-Based Similarity Measurement

Through the conversion of trajectories, a matrix with grid-codes is generated to represent a trajectory. The trajectory similarity calculation can be done by comparing and analyzing codes in the two matrices. In this study, there are mainly two types of spatial similarity measures. These are the common-locations similarity, which consider the common locations visited by the two trajectories, and the structural similarity, which considers the self-intersection of a trajectory. 

For the first category, the similarity of the common location of the trajectory mainly focuses on the common locations a moving object has visited. This may be useful for location recommendation services, or for finding objects which move through certain points of common interest (e.g., emergency locations, terrorist locations, etc.). 

However, in many location-based applications it is not enough to consider only the common locations of two trajectories. Figure 5 shows an example with two taxi trajectories in Shanghai. The taxi activities are considerably restricted by geographical space, and the range of the two trajectories varies a lot. Since both trajectories in Figure 5 include the route to the airport, if only the common locations are considered the similarity between the two trajectories will be high. This result is obviously unreliable in the analysis of behavior pattern. Consequently, it is very important to consider repetitive and self-intersecting features of trajectories in the applications of movement patterns (e.g., analysis of travel behavior preference, hotspot extraction). As for the second category, the structural similarity algorithm has been designed. It not only focuses on the common points of interest that trajectories pass by, but also on the information of repetitive active regions and self-intersections of trajectories.

The following sections explain the measurements in detail.

#### 2.3.1. Similarity of Common Locations

A spatial similarity measurement (*Sim_loc_* (*T_i_*,*T_j_*)), focusing on common locations in trajectories, is introduced in this subsection. It counts the number of the common locations and calculates its percentage of all location points in trajectories as the similarity of the two trajectories. Note that in the process of calculating the similarity of common location points, we focus on whether a location point is passed or not, regardless of how many times it passes. Therefore, before calculating the similarity of common locations, we need to do unique value processing for each track.

Let *T_i_* and *T_j_* be two trajectories. After trajectory conversion, *T_i_* and *T_j_* are converted into two trajectory matrices (*T_ matrix_i_*, *T_matrix_j_*). The spatial similarity measure between these two trajectories is defined based on the common elements in these trajectory matrices.
(7)$$Matrix_{i}=f(T\_matrix_{i}),$$
(8)$$Matrix_{common}=Matrix_{i}\cap Matrix_{j},$$
(9)$$Matrix_{total}=Matrix_{i}\cup Matrix_{j},$$
(10)$$Sim_{loc}\left(T_{i},T_{j}\right)=\frac{sum(Matrix_{common})}{sum(Matrix_{total})}.$$
where *f* (***) represents a function of the unique value calculation; *Matrix_i_* is the grid-coded trajectory matrix after the unique value calculation; *Matrix_common_* is the matrix of the common locations; *Matrix_total_* is the matrix of the total locations of the trajectory *i*, and *j*. *Sim_loc_* (*T_i_*,*T_j_*) is the ratio of the sum of the two matrix values.

Please note that the above similarity measure satisfies the following properties:*Sim_loc_* (*T_i_*,*T_j_*) >= 0.*Sim_loc_* (*T_i_,T_j_*) = *Sim_loc_* (*T_j_,T_i_*).*Sim_loc_* (*T_i_*,*T_j_*) belongs to [0,1].

In Figure 6, the similarity calculation of the taxi trajectories *T1* and *T2* is visualized. After the trajectory conversion and the unique value processing, trajectory matrices are obtained and displayed in the grid, where 1 indicates that the trajectory passed the grid and 0 indicates that it did not pass. After calculation, *T1* passes through 27 grid cells, *T2* passes through 24 grid cells. The intersection of trajectory matrices (see the common part) is 18, and the union is 27 + 24 − 18 = 33. So, the result of the similarity is 18/33 = 0.55.

#### 2.3.2. Structural Similarity Measurement in Trajectories

The structural similarity measurement (*Sim_str_* (*T_i_*,*T_j_*)) focuses on the information of repetitive active regions and self-intersections in trajectories. It mainly focuses on the number of visits in the common locations of the trajectories and can help to extract the behavior preferences of moving objects in the analysis of movement patterns. The structural similarity of the two trajectories is defined as:(11)$$Matrix_{common}=\mathrm{minimum}(Matrix_{i},Matrix_{j}),$$
(12)$$Matrix_{total}=Matrix_{i}+Matrix_{j}-Matrix_{common},$$
(13)$$Sim_{str}\left(T_{i},T_{j}\right)=\frac{sum(Matrix_{common})}{sum(Matrix_{total})},$$
where *Matrix_i_* is a trajectory matrix whose values are the number of times the grid cell has been visited. *Matrix_common_* is a matrix whose values represent the common times of the trajectory *i* and *j* to visit this grid cell. *Matrix_total_* is the union matrix of two matrix, and its values are obtained by subtracting the minimum visited times from the total visited times of the trajectory *i*, and *i*. *Sim_str_* (*T_i_*,*T_j_*) is the ratio of the sum of the two matrix values.

Please note that the above similarity measure satisfies the following properties:*Sim_str_* (*T_i_*,*T_j_*) >= 0.*Sim_str_* (*T_i_,T_j_*) = *Sim_str_* (*T_j_,T_i_*).*Sim_str_* (*T_i_*,*T_j_*) belongs to [0,1].

Using this algorithm, the process of calculating the structural similarity of trajectories *T1* and *T2* is shown in Figure 7. The attribute value represents the number of times the trajectory passes through the grid cell. For *T1* and *T2*, the sum of the grid attributes of the common part is 21, and the sum of these two trajectory matrices are 34 and 32, respectively. So, the result of the similarity is 21/(34 + 32 − 21) = 0.47.

#### 2.3.3. The Combined Spatial Similarity

There are some applications that need to consider these two similarities comprehensively. For example, taxi trajectories data, as a spatial trajectory data for profit, are mainly distributed in the down areas with a lot of pick-up/drop-off points. By calculating the structural similarity of the trajectories, a similarity set with a large number of trajectories can be extracted, which is mainly distributed in the city center. In order to explore the travel patterns within the similar set, it is necessary to refine the trajectory using the similarity of the common locations. For spatial similarity analysis of this type of application, a comprehensive metric is needed. It can be used as the distance measure to subdivide similar sets and further mine their spatial characteristics.

Xia et al. [41] considered that spatial and temporal characteristics have the same weight in the spatiotemporal similarity calculation of trajectories. Abraham et al. [42] transformed the trajectory into binary code and believe that location, sequence and other factors have the same effect on the combined similarity. Consequently, in the process of calculating the combined spatial similarity, our research considers that the common location similarity and the structural similarity have the same weight. The combined spatial similarity measure is obtained by:(14)$$Sim\left(T_{i},T_{j}\right)=(Sim_{loc}+Sim_{str})/2,$$

The result also satisfies the following properties:*Sim* (*T_i_*,*T_j_*) >= 0.*Sim* (*T_i_*,*T_j_*) = *Sim* (*T_j_*,*T_i_*).*Sim* (*T_i_*,*T_j_*) belongs to [0,1].

In the end, the similarity of *T1* and *T2* is (0.55 + 0.47)/2 = 0.51.

Note that each kind of similarity measurement algorithm can be used separately according to different application scenarios and needs. The applicable scenarios have been illustrated in the corresponding subsections.

## 3. Experiments

We have implemented our method and conducted an extensive set of experimental studies in order to (1) test the proposed techniques; and (2) compare our method with another model. In this section, a case study is conducted using taxi trajectory data from Shanghai, China (all administrative districts except Chongming district). Chongming District is a suburban county in Shanghai far away from the city area. The taxis in this area are restricted by the natural environment, resulting in rare contact with the mainland of Shanghai. According to the statistics of our floating car data, less than 1% of the data is related to Chongming District. Since this section is to verify and evaluate our algorithm, Shanghai city area with more trajectories was selected as the research area. The method used is the combined spatial similarity algorithm proposed in our research. The experiments were conducted on an Intel Core i5-7500 Quad-core machine running Windows 10 with 16 GB of RAM and a 250 GB SATA2 512-MB hard drive.

### 3.1. Data Preprocessing and Experimental Setup

#### 3.1.1. The Trajectory Data Description

The taxi trajectory data used in this paper were provided by a commercial company in Shanghai. They are temporally ordered position records collected from about 6000 Global Positioning System (GPS) enabled taxis. The data was collected over a period of 7 days from 4 June to 10 June 2018. The average sampling interval of the data was 10 s. In the database, the trajectory data of each vehicle is a series of position records arranged in chronological order. Each record has many attributes, i.e., Taxi identifier Time, Speed, Direction, Current location (longitude, latitude) and Passenger state. The “Taxi identifier” is a unique identifier of a taxi, while “Time” contains an accurate date and time for each record. “Speed” represents the speed of a taxi at a given time. “Direction” is the horizontal angle measured clockwise from the north direction. “Passenger state” is a Boolean variable that denotes whether the taxi is carrying passengers or not. In space, the raw GPS trajectory data is represented by a series of discrete points. Figure 8 illustrates the sample points of one taxi with the identification number of 10,383 on the 4 June 2018. The sample points are red, while the trajectory route is green. The trajectory route is generated by sample points in chronological order.

#### 3.1.2. Grid Generation

In our example, the road network of Shanghai was used to determine the size of the grid cell, and subsequently utilized to generate the grid for Shanghai. The roads were obtained from OpenStreetMap (OSM) (http://www.openstreetmap.org) because it is freely available [54] and has comparable quality to the authority data based on our local knowledge and the existing study [55].

The roads in Shanghai can be divided into five levels. These are expressways, main roads, trunk roads, secondary trunk roads and branch roads. In this study, the method of visual interpretation was adopted to select the suitable road network. For example, in the blue frame (Figure 9), the branch road network contains too many small paths, which can divide the “Town God’s Temple” area into many fragmented patches. Consequently, the second trunk and above road (i.e., expect branch roads) should be selected as input data.

The spatial distribution of the road network in Shanghai is unbalanced (as shown in Figure 10a,b). The density distribution of the road network coincides with the ring road (Figure 10b), which are the two ring expressways. The road network density in the inner ring area is the highest, successively, in the area between the inner and outer rings and in the outer ring area. Therefore, the study area was divided into three sub regions. These are the inner ring area, the area between the inner and outer rings, and the outer ring area. The road network spacing per unit area of the three regions was calculated, respectively. As shown in the histogram (Figure 10c), the average distance of the road spacing in different zones varied a lot. The average distance in the inner ring area is 350 m (mainly ranging from 200 to 450 m), followed by the 700 m in the area between the inner and outer rings (mainly ranging from 400 to 1000 m) and 1500 m in the outer ring area (mainly ranging from 800 to 2000 m). Consequently, the grid size of the three sub areas is 350, 700 and 1500, respectively (Figure 10d).

### 3.2. Effectiveness of the Method

In order to evaluate the effectiveness of our approaches, we randomly selected 2000 taxi trajectories (i.e., 1000 pairs) from our dataset. Each trajectory contained at least 200 sampling points. By visualizing the trajectories on the digital map, similar trajectory pairs were manually labeled as the ground truth. A total of 72 trajectories were labeled in this experiment. The precision and recall of the similar pairs in different similarity threshold were used in this section. The precision is the portion of real similar pairs (as indicated by the ground truth) in all similar trajectory pairs found by the method in our study (i.e., Precision = Real similar trajectories/Similar trajectories). The recall is the ratio of the number of the real similar trajectory pairs in the results calculated in this case to the number of real similar trajectory pairs in the ground truth (i.e., Recall = Real similar trajectories/ground truth).

The Table 1 shows the results of detecting similar pairs of trajectories with different similarity thresholds, where the term of Real Similar Trajectories denotes the similar pairs correctly identified in our method. With the increase of the Similarity Threshold (from 0.6 to 0.8), the precision was significantly improved, while the recall decreased slightly. This is because the higher the similarity threshold, the fewer similar trajectories can be found, which reduces the denominator of precision and the numerator of recall. As shown in Table 1, when the similarity threshold is greater than 0.8, the precision and recall are 0.969 and 0.861, respectively. This performance of our approach can meet the needs of data mining, especially for trajectories with different geographic coordinates and shapes. As shown in Figure 11, the similarity of the two trajectories (i.e., T1 and T2) is 0.85. The exact geographic coordinates and the shape of the two trajectories (T1 and T2) are different, but the main places that the two trajectories passed through are consistent (e.g., the area of Shanghai General Hospital, Yu Garden). Therefore, the two trajectories are considered to be similar.

As shown in Figure 11a, these sampling points of the two trajectories belong to the same area, Shanghai General Hospital. However, due to the different locations of the pick-up points, the geographic coordinates and shapes of the two trajectories in this area are different. The area in Figure 11c is a very famous scenic spot named Yu Garden, where the density of vehicles and population is high. Vehicles are likely to choose other routes to reach this area due to the road congestion. In our approach, we focus on the place where the trajectory passes rather than the geographic coordinates. Therefore, the trajectories to reach this area are considered similar. Additionally, due to the unstable GPS signal, the geographic coordinates of the sampling points will have an error of 10−20 m. The shape and distance between sequences of the same route may be different (Figure 11b), which may cause the calculation results to be inaccurate.

In all, unstable GPS signal, vehicle lane change, traffic congestion and the past travel experience can lead to different trajectory shapes and road information. However, the main locations/places they pass through are consistent, so the two trajectories should be considered very similar. In our method, shape details and the road information of trajectories are neglected while emphasizing the integrity of the trajectory. Consequently, the algorithm proposed in this paper has good robustness.

According to this feature, the trajectory similarity measurement proposed in this paper can be well applied in location recommendation systems or the group analysis mentioned in the introduction. This is because the calculation of trajectories’ similarity in these methods mainly pay attention to whether the trajectory reaches a specific area. In addition, based on the similarity results calculated in this paper, typical routes can be extracted from the mass trajectories to provide guidance for travel route recommendations and route planning of urban public transportation systems. For verification, we extracted the trajectories of Pudong International Airport and calculated their similarities. Then, as a traditional density-based clustering algorithm, the Density-Based Spatial Clustering of Applications with Noise (DBSCAN) algorithm was used to visualize our similarity calculation results. The DBSCAN algorithm requires two parameters: *ε* (*eps*), and the minimum number of points required to form a high-density area (*minPts*) [56]. It starts with an arbitrary unvisited point and then explores the *ε*-neighborhood of this point. If there are enough points in the *ε*-neighborhood, a new cluster is established, otherwise the point is labeled as noise.

Figure 12a illustrates the 947 visit trips on 4 June 2018 (excluding the departure trajectories from the airport). After calculating the similarity of trajectories, the DBSCAN algorithm was used to cluster the airport trips, but the distance parameter in the DBSCAN algorithm was replaced by trajectory similarity. In the DBSCAN algorithm, the parameters were set as *esp* = 0.8, *minpts* = 10, i.e., the similarity of trajectories is more than 0.8, and the number of trajectories in the ε-neighborhood is more than 10. The DBSCAN algorithm can remove some noise trajectories with low similarity and divide airport trajectories into nine categories based on parameter settings. The visualization results are shown in the Figure 12b. By utilizing the similarity results calculated in our study, meaningful trajectory clusters and typical routes can be generated.

### 3.3. Comparison with State-of-the-Art Model

The performance of our model is evaluated by comparing it with Fast Dynamic Time Warping (FastDTW), which is an approximate Dynamic Time Warping (DTW) algorithm that provides optimal or near-optimal alignments with an O(N) time and memory complexity [57]. The DTW algorithm is widely used in similarity calculations and its performance has been verified by several authors [38,58,59]. However, the quadratic time and space complexity of DTW is limited to small time series datasets. FastDTW can, on the one hand, be run on much larger data sets. It is also an order of magnitude faster than DTW. Consequently, the FastDTW algorithm is used for model efficiency comparison experiments, including data storage and the elapsed time of the similarity calculation. The experiments were conducted by using the same data set with opensource codes FastDTW https://pypi.org/project/fastdtw/ and DTW https://pypi.org/project/dtw/.

We selected 10 sets of data for the experiment, and the number of trajectories was 50, 100, …, 500, respectively. For the data storage, not only the sizes of data space but also the number of trajectory records were compared between the original trajectory and the converted trajectory. As shown in the Figure 13, the data space for original and converted data changed linearly with the increased number of trajectories. The comparison shows a significant difference between the original trajectory and the converted trajectory. The original trajectory occupies 34-times more storage space than the converted. By transforming and converting the trajectory data, the record of the data is only 1/10 of the original data.

Subsequently, the performance of the similarity search technique between our algorithm and FastDTW algorithm is measured by comparing the average elapsed time. As shown in Figure 14, the experimental results confirm that the elapsed time of the two models increases linearly as the number of trajectories increase. However, the time consumption for the FastDTW algorithm is higher than our method (about 2.4 times). There are two main reasons. First, although our model needs a part of time for trajectory conversion, trajectory conversion can greatly reduce the amount of data to reduce the elapsed time. Second, our model does not need to calculate the distance between each pair of sampling points. Therefore, our proposed similarity algorithm (SGCD) has great advantage in processing large-scale trajectory data.

In addition, we use the FastDTW model to calculate the similarity of 1000 pairs of trajectories used in Section 3.2. The result of the FastDTW model is the sum of the shortest distance between aligned sampling points. The smaller the distance between the trajectories, the more similar they are. For each distance function d, we can define its associated similarity function as *s*(*x*, *y*) = 1/(1 + *d* (*x*, *y*)), which ranges from 0–1. Note that property should be dropped to avoid dimensional problems [60,61]. 

The calculation results are shown in Table 2. With the increase of the similarity threshold, the precision of the FastDTW model is acceptable, but the data recall rate and the number of correct trajectories identified are very low. There are two main reasons. First, it is related to the transformation of distance function and similarity function in the calculation process, and the outliers have a greater impact on the calculation results. Secondly, the structural similarity of trajectories is considered in our model, and the principle of the two models and the definition of similar trajectories are different. Therefore, the FastDTW model is not suitable for the location-based trajectory similarity research mentioned in this study.

## 4. Conclusions and Future Work

In this paper, an algorithm of Spatial Grid Coding Distance (SGCD) was designed to calculate the trajectories’ similarity. Instead of relying on aligned sample points as in traditional approaches, it can use the grid to convert the trajectory data and identify the similar trajectories passing through the same places. In order to obtain better performance and acceptable accuracy, a rule for determining the appropriate grid size is developed by calculating the network spacing. By considering the self-intersection characteristics of trajectories, two similarity calculation algorithms are designed: namely, the common-locations similarity and structural similarity. Experimental study on real datasets verified the advantages and efficiency of our algorithm.

The similarity calculation of vehicle trajectory can be used in many traffic-related applications. For example, in traffic congestion recognition applications, our algorithm can be adjusted by considering the number of consecutive sampling points in each grid and subsequently utilizing them to identify the congestion area. In addition, in the field of human sociology, its application prospect is also widely concerned (e.g., behavior pattern analysis, service sharing). Note that taxis are one of the important ways to travel. If data for all types of travel could be obtained (e.g., private cars), more valuable information will be mined. For example, the movement of private cars is more regular than that of taxis. In general, the main activity on weekdays is commuting (especially at specific times of the day). By calculating the common location similarity and structural similarity of the trajectory, information such as the main active area and routes of the trajectory can be extracted more quickly and accurately. This would be helpful for the development of behavior pattern analysis and carsharing services. As a continuation, we plan to use the algorithm of similarity measurement to extract semantic information from different types of trajectories and further explore human behavior patterns. Some other data, such as weather factors, regional terrain factors and social media data, may be added to improve the accuracy of the results. 

## Figures and Tables

**Figure 1 sensors-20-03118-f001:**
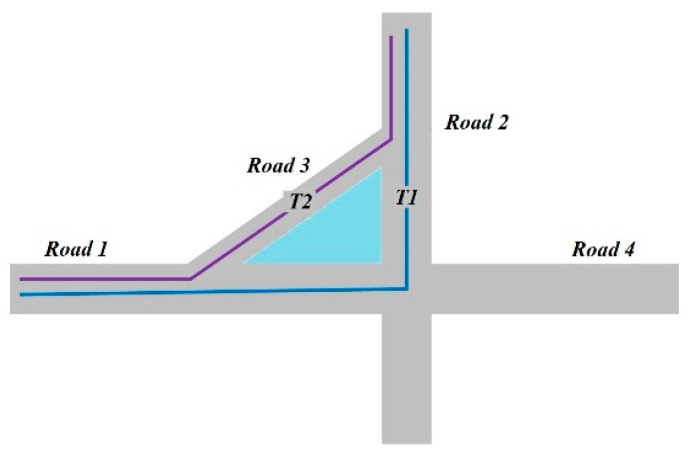
Example of the movement of two vehicles on different road segment.

**Figure 2 sensors-20-03118-f002:**
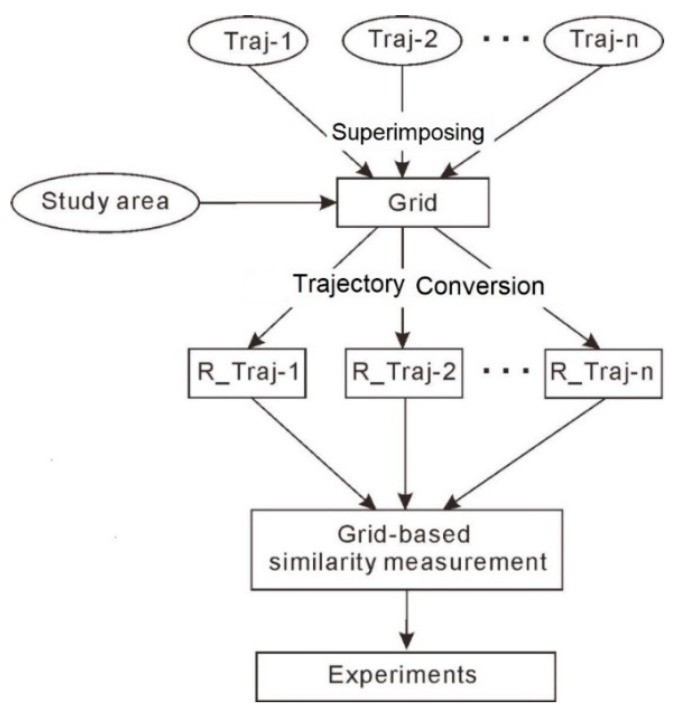
Workflow of the similarity measurement.

**Figure 3 sensors-20-03118-f003:**
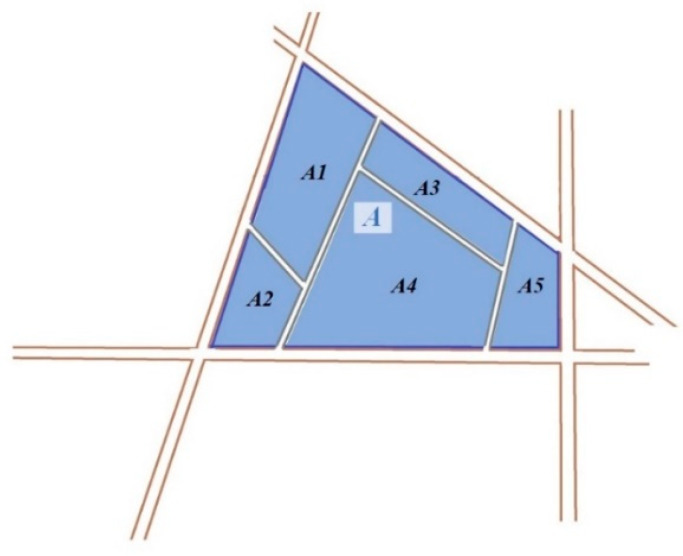
Example of branch road in a garden.

**Figure 4 sensors-20-03118-f004:**
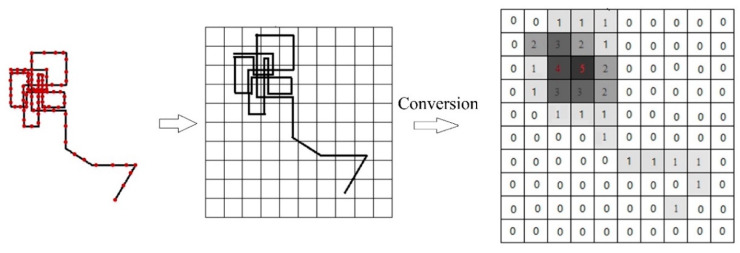
An example of trajectory conversion.

**Figure 5 sensors-20-03118-f005:**
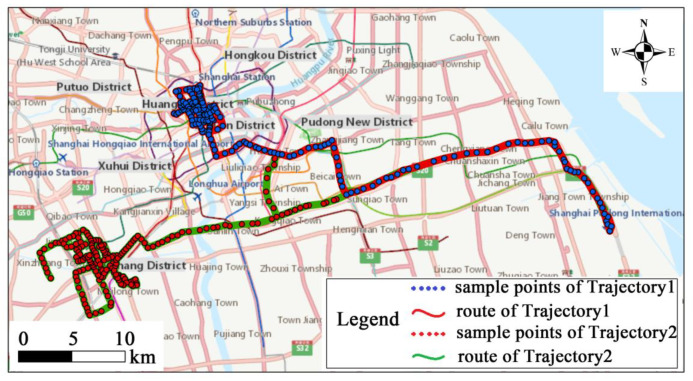
The trajectories of two taxis.

**Figure 6 sensors-20-03118-f006:**
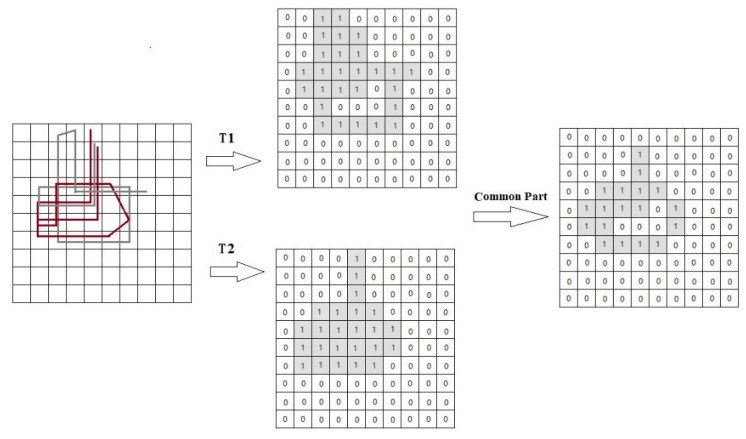
Example of common locations similarity measurement.

**Figure 7 sensors-20-03118-f007:**
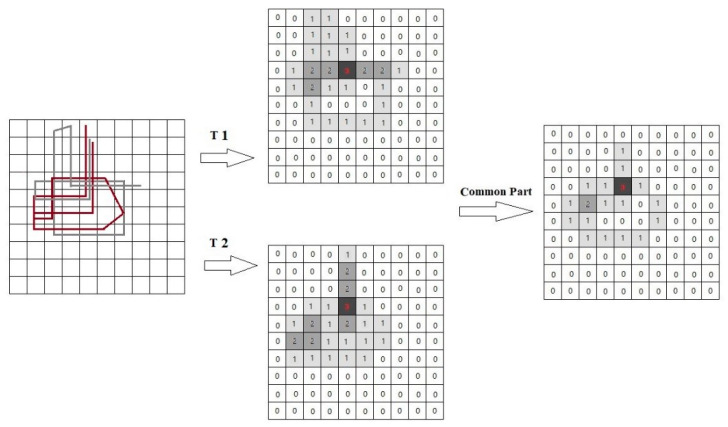
Example of structural similarity measurement.

**Figure 8 sensors-20-03118-f008:**
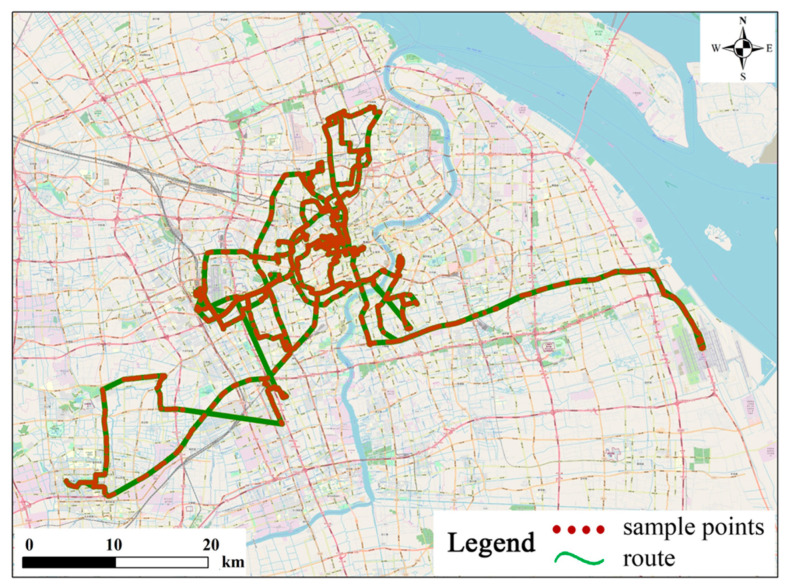
The trajectory of a taxi with a taxi identifier of 10,383 on the 4 June 2018.

**Figure 9 sensors-20-03118-f009:**
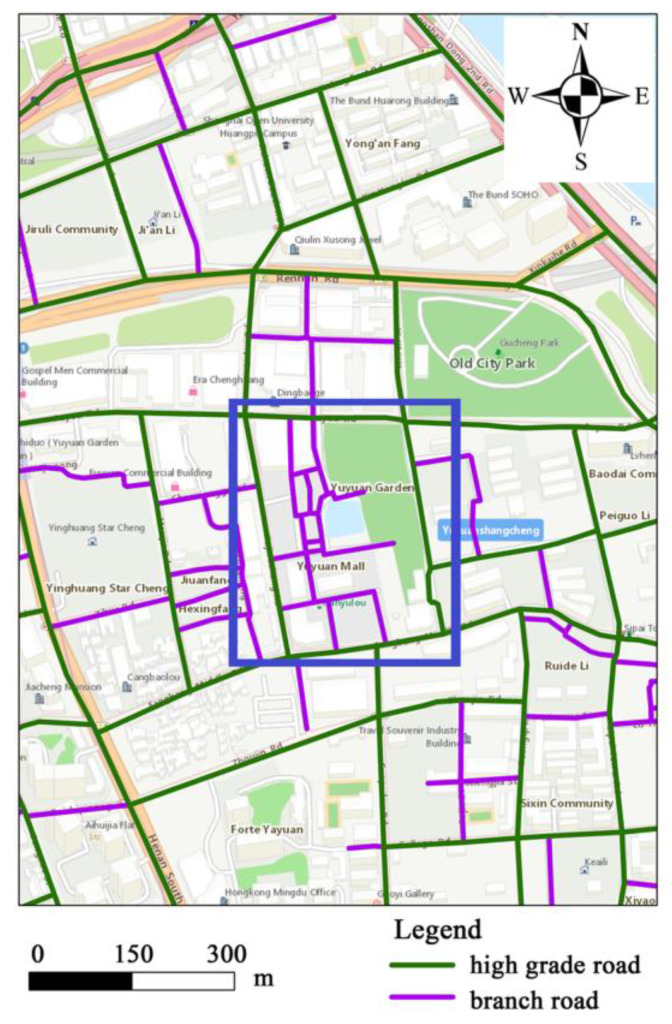
Visualization of different grade road network.

**Figure 10 sensors-20-03118-f010:**
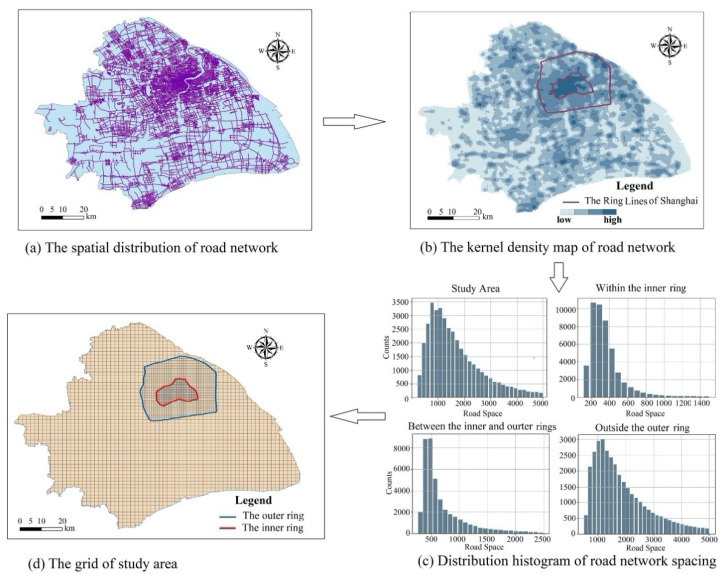
Grid size determination procedure.

**Figure 11 sensors-20-03118-f011:**
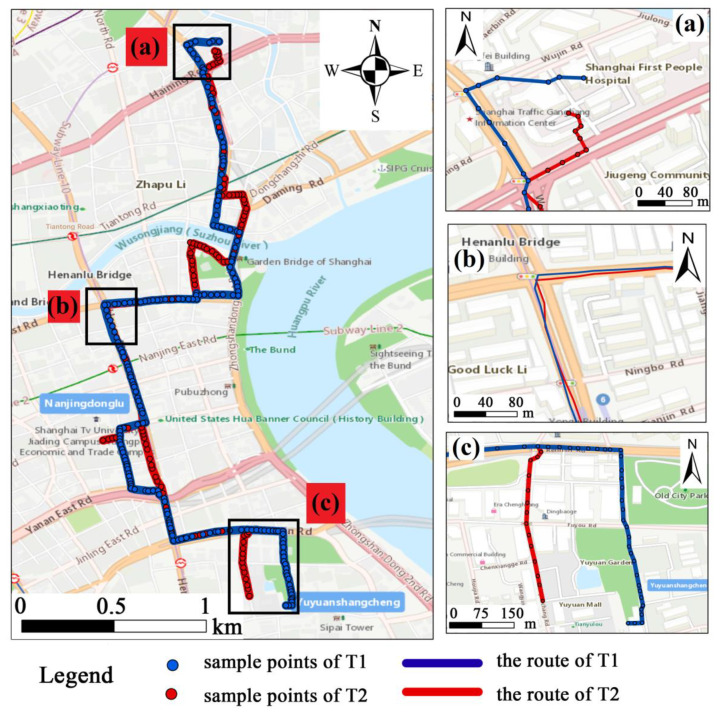
Example of two similar trajectories.

**Figure 12 sensors-20-03118-f012:**
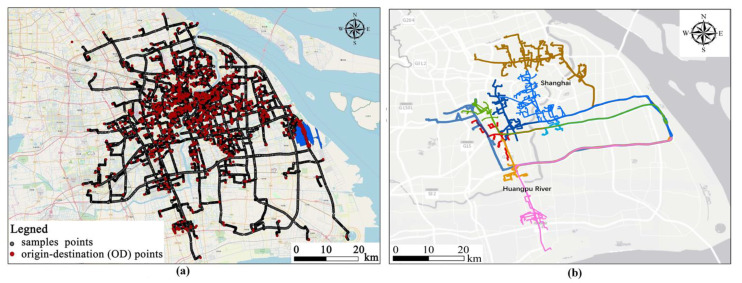
(**a**) Airport visit trips of Pudong international airport on 4 June 2018. (**b**) Clustering results of Pudong airport travel based on similarity. The nine colored lines indicate the categories divided by DBSCSN based on trajectories similarity.

**Figure 13 sensors-20-03118-f013:**
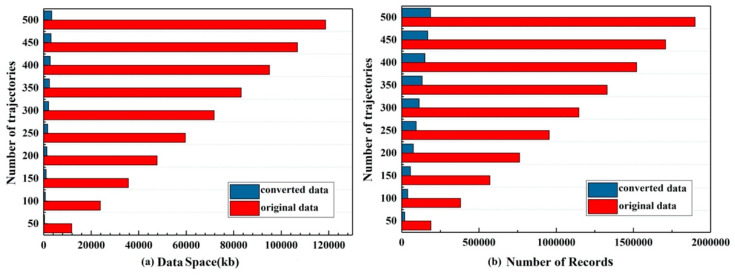
Comparison of the (**a**) data space (kb) and (**b**) the number records for the trajectory dataset.

**Figure 14 sensors-20-03118-f014:**
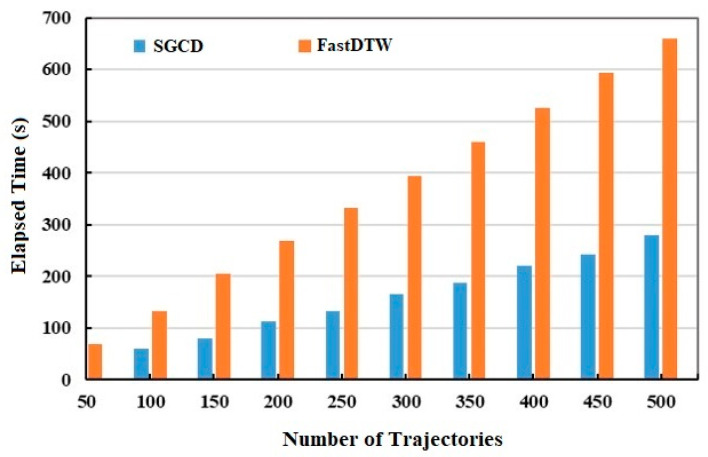
Comparison of the elapsed times for the trajectories.

**Table 1 sensors-20-03118-t001:** The results of similar pairs detecting.

Similarity Threshold	Similar Trajectories	Real Similar Trajectories	Precision	Recall
0.6	214	72	0.336	1
0.7	130	70	0.538	0.972
0.8	64	62	0.969	0.861

**Table 2 sensors-20-03118-t002:** The results of similar pairs detecting by Fast Dynamic Time Warping (FastDTW).

Similarity Threshold	Similar Trajectories	Real Similar Trajectories	Precision	Recall
0.6	26	20	0.769	0.278
0.7	20	16	0.80	0.222
0.8	14	10	0.714	0.139

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
