# Peer review of "A Grid-Based Approach for Measuring Similarities of Taxi Trajectories"

_sensors, 2020, doi:10.3390/s20113118_

Round 1
Reviewer 1 Report
Review
The paper presents an algorithm of Spatial Grid Coding Distance (SGCD) is designed to calculate the trajectories similarity. To convert the trajectory data and identify the similar trajectories passing through same places was used the grid.
The paper is interesting and well written, but a few things need clarification.
- Line 184
“A raster data of road network density is generated by kernel density calculation, and its attribute value is the road network density within the range of cells. The cell can be called a density unit in our research.”
The size of the grid cell depends on the density of traffic. Traffic density is different on weekends than on working days. Is there a different grid for the same area for working days, weekends and holidays?
What will the structural similarity of the trajectory look like then?
- Table 1 requires clarification - how many pairs of 1000 tested were correctly identified? What results of recognizing the same trajectories does the FastDTW method give?
Minor
There is no date in references 25, 27, 36 and 40.
Author Response
Reviewer #1:
The paper presents an algorithm of Spatial Grid Coding Distance (SGCD) is designed to calculate the trajectories similarity. To convert the trajectory data and identify the similar trajectories passing through same places was used the grid. The paper is interesting and well written, but a few things need clarification.
Reply:
Thank you very much for your valuable comments. We have revised these contents and improved the manuscript in accordance with your advices and suggestions.
Point 1: Line 184 “A raster data of road network density is generated by kernel density calculation, and its attribute value is the road network density within the range of cells. The cell can be called a density unit in our research.”
The size of the grid cell depends on the density of traffic. Traffic density is different on weekends than on working days. Is there a different grid for the same area for working days, weekends and holidays?
What will the structural similarity of the trajectory look like then?
Reply:
In the proposed method, the determination of grid size is based on the density of static road network rather than the density of dynamic traffic. The reason has been stated in Section 2.1.1, such as “Considering that the road network in a city is an important indicator of urban development level [45]. Developed areas normally have denser roads and more points of interest (e.g., hospitals, schools), while less developed areas have a sparse road network and fewer meaningful places. Since our research focuses on the spatial similarity calculation based on common locations /places, road network spacing in the city is used as the basis for sizing the grid cells.” Consequently, we used the same grid for the same area on weekdays and weekends. Also, we do not consider the influence of traffic density in the process of structural similarity of the trajectory. We can pay attention to this difference of traffic density in the process of analyzing specific problems (e.g., behavior pattern analysis) by using trajectory similarity in the future.
Point 2: Table 1 requires clarification - how many pairs of 1000 tested were correctly identified? What results of recognizing the same trajectories does the FastDTW method give?
Reply:
In the revised version, we have changed Table 1 according to the suggestion. A detailed description of the result has been added in the revised version (Lines 400-406).
“Table 1. The results of similar pairs detecting
|
Similarity threshold |
Similar trajectories |
Real similar trajectories |
Precision |
Recall |
|
0.6 |
214 |
72 |
0.336 |
1 |
|
0.7 |
130 |
70 |
0.538 |
0.972 |
|
0.8 |
64 |
62 |
0.969 |
0.861 |
The Table 1 shows the results of detecting similar pairs of trajectories with different similarity thresholds. Whereas the term of Real similar trajectories denotes the similar pairs correctly identified in our method. With the increase of the similarity threshold (from 0.6 to 0.8), the precision is significantly improved, while the recall decreases slightly. This is because the higher the similarity threshold, the fewer similar trajectories can be found, which reduces the denominator of precision and the numerator of recall.”
Regarding the suggestion of the result of FastDTW, because the model comparison is in section 3.3, we added it in the appropriate position in the revised manuscript (Lines 486-499).
“In addition, we use the FastDTW model to calculate the similarity of 1000 pairs of trajectories used in Section 3.2. The result of the FastDTW model is the sum of the shortest distance between aligned sampling points. The smaller the distance between the trajectories, the more similar they are. For each distance function d, we can define its associated similarity function as ?(?, ?)=1/(1+? (?, ?)), which ranges from 0-1. Note that, property should be dropped to avoid dimensional problems [60,61].
The calculation results are shown in Table 2. With the increase of the similarity threshold, the precision of the FastDTW model is acceptable, but the data recall rate and the number of correct trajectories identified are very low. There are two main reasons. First, it is related to the transformation of distance function and similarity function in the calculation process, and the outliers have a greater impact on the calculation results. Secondly, the structural similarity of trajectories is considered in our model, and the principle of the two models and the definition of similar trajectories are different. Therefore, the FastDTW model is not suitable for the location-based trajectory similarity research mentioned in this study.
Table 2. The results of similar pairs detecting by FastDTW ”
|
Similarity threshold |
Similar trajectories |
Real similar trajectories |
Precision |
Recall |
|
0.6 |
26 |
20 |
0.769 |
0.278 |
|
0.7 |
20 |
16 |
0.80 |
0.222 |
|
0.8 |
14 |
10 |
0.714 |
0.139 |
Reference (Lines 673-675)
- Ontañón, S. An overview of distance and similarity functions for structured data. Artificial Intelligence Review 2020, 1-43.
- Tversky, A. Features of similarity. Psychological review 1977, 84, 327.
Point 3: There is no date in references 25, 27, 36 and 40.
Reply:
We have added date it in the revised manuscript.
Reviewer 2 Report
I find your study as an interesting contribution to the measuring similarities of vehicle trajectories.
However I would like to list a few minor remarks or questions to the authors - by dispelling my doubts you will improve and supplement your paper.
lines 168-171 – Are there any circumstances except of the monocentric and polycentric structure? Can you give any examples proving the lower network density in suburban areas? In my opinion a proper reference is recommended in this part.
lines 329 - 330 – What was the reason for excluding this one district? – it is not so obvious for a foreigner.
lines 353 – 354 – You have chosen the OpenStreetMap as a source of data for the road network - what are the advantages of this data source compared to other sources of the city's road network?
line 362 – “shanghai” without capital letter
lines 479 – 485 – Can you supplement your “Conclusions” part by explaining how congestion (at specific times of the day) and/or the type of vehicle may affect your further studies? There are very narrow roads - in particular in the strict, historic centres of towns and cities in European or African regions – will you consider road ‘widths’ are usefulness for traditional taxis?
General remark - instead of descriptions before or below figures, you can add legends - in some cases (in particular - maps) they would be more appropriate
Author Response
Reviewer #2:
I find your study as an interesting contribution to the measuring similarities of vehicle trajectories. However, I would like to list a few minor remarks or questions to the authors - by dispelling my doubts you will improve and supplement your paper.
Reply:
Thank you very much for your valuable comments. We have revised these contents and improved the manuscript in accordance with your advices and suggestions.
Point 1: lines 168-171 – Are there any circumstances except of the monocentric and polycentric structure? Can you give any examples proving the lower network density in suburban areas? In my opinion a proper reference is recommended in this part.
Reply:
There are three main theories of urban spatial structure, namely concentric circle model, sector theory and multi-core theory. In terms of the number of urban centers, the urban spatial structure has two types: monocentric and polycentric structure.
Some references and detailed description about the spatial distribution of road network have been added in the revised manuscript (Lines 168-175).
“The second problem is to solve the imbalanced spatial distribution of the road network. Many studies have shown that urban road network density is closely related to spatial structure [46,47]. Cai et al. used the kernel density estimation to assess the spatial pattern of road density and clearly reflected that the kernel densities of roads decreased with greater distance from urban core areas or central business districts [48]. However, not all cities have only one central business district. There are three main theories of urban spatial structure, namely concentric circle model [49], sector theory [50] and multi-core theory [51]. In terms of the number of urban centers, the urban spatial structure has two types: monocentric and polycentric structure.”
Reference (Lines 643-655):
- Iacono, M.; Levinson, D. Mutual causality in road network growth and economic development. Transport Policy 2016, 45, 209-217.
- Sreelekha, M.; Krishnamurthy, K.; Anjaneyulu, M. Interaction between road network connectivity and spatial pattern. Procedia technology 2016, 24, 131-139.
- Cai, X.; Wu, Z.; Cheng, J. Using kernel density estimation to assess the spatial pattern of road density and its impact on landscape fragmentation. International Journal of Geographical Information Science 2013, 27, 222-230.
- Burgess, E. The growth of the city in Park, RE, Burgess EW and McKenzie RD (eds), The city. Chicago: University of Chicago Press: 1925.
- Hoyt, H. The structure and growth of residential neighborhoods in American cities; US Government Printing Office: 1939.
- Harris, C.D.; Ullman, E.L. The nature of cities. The Annals of the American Academy of Political and Social Science 1945, 242, 7-17.
Point 2: lines 329 - 330 – What was the reason for excluding this one district? – it is not so obvious for a foreigner.
Reply:
We have added the detail description in the revised manuscript (Lines 333-340).
“In this section, a case study is conducted using taxi trajectory data from Shanghai, China (All administrative districts except Chongming district. Chongming District is a suburban county in Shanghai far away from the mainland. The taxis in this area are restricted by the natural environment, resulting in rare contact with the main districts of Shanghai. According to the statistics of our floating car data, less than 1% of the data is related to Chongming District. Since this section is to verify and evaluate our algorithm, Shanghai mainland with more trajectories is selected as the research area.).”
Point 3: lines 353 – 354 – You have chosen the OpenStreetMap as a source of data for the road network - what are the advantages of this data source compared to other sources of the city's road network?
Reply:
A detailed description of the advantages has been added in the revised manuscript (Lines 362-366)
“The roads are obtained from OpenStreetMap (OSM) (http://www.openstreetmap.org), which has two main advantages: First, OSM is open data. When geographic information availability and access are restricted, it will become an alternative source of data [54]; Second, The quality of this geo-data is recognized and has become the foundation of some scientific publications in many research fields.[55]”
Reference (Lines 659-663):
- Watanabe, T.; Yamaguchi, T.; Koda, S.; Minatani, K. Tactile map automated creation system using openstreetmap. In Proceedings of International Conference on Computers for Handicapped Persons 2014, pp. 42-49.
- Fan, H.; Zipf, A.; Fu, Q.; Neis, P. Quality assessment for building footprints data on OpenStreetMap. International Journal of Geographical Information Science 2014, 28, 700-719.
Point 4: line 362 – “shanghai” without capital letter
Reply:
We have corrected it in the revised manuscript (Line 375)
Point 5: lines 479 – 485 – Can you supplement your “Conclusions” part by explaining how congestion (at specific times of the day) and/or the type of vehicle may affect your further studies? There are very narrow roads - in particular in the strict, historic centres of towns and cities in European or African regions – will you consider road ‘widths’ are usefulness for traditional taxis?
Reply:
For the first suggestion, this part has been revised as follow (Lines 509-523).
“The similarity calculation of vehicle trajectory can be used in many traffic related applications. For example, in traffic congestion recognition applications, our algorithm can be adjusted by considering the number of consecutive sampling points in each grid, and subsequently utilized to identify the congestion area. In addition, in the field of human sociology, its application prospect is also widely concerned (e.g., behavior pattern analysis, service sharing). Note that, taxi is one of the important ways to travel. If data for all types of travel could be obtained (e.g., private cars), more valuable information will be mined. For example, the movement of private cars is more regular than that of taxis. In general, the main activity on weekdays is commuting (especially at specific times of the day). By calculating the common location similarity and structural similarity of the trajectory, information such as the main active area and routes of the trajectory can be extracted more quickly and accurately. This would be helpful for the development of behavior pattern analysis and carsharing services. As a continuation, we plan to use the algorithm of similarity measurement to extract semantic information from different type of trajectories and further explore human behavior patterns. And some other data, such as weather factors, regional terrain factors and social media data, may be added to improve the accuracy of the results.”
For another question:
The width of the road was not considered in the process of setting the grid size. There are two reasons. First, in the OSM data, we can only extract information such as the level and length of the road, excluding the width information. The information of road network level and length help us to determine the grid size when solving practical problems. For example, in the calculation of taxi trajectory similarity, we select the level of second trunk and above road network to determine the grid size, as described in Section 3.1. Second, according to the classification of road level, road capacity is the main basis for judgment. In addition, the tasks and functions of road use are also important factors in judging the level of the road network (Jones, 1986). If the road is more important (i.e. traffic flow is large), even if the road in the historic center of town is narrow, its road network level will be set high. For example, the main urban area of Rome will also be distributed with trunk roads. Therefore, the determination of road level is affected by many factors, not entirely determined by the width of the road. Even the roads in the historical districts with very narrow width mentioned will not affect the application of our algorithm and the use of taxis as long as the road traffic flow meets the demand.
Jones I D. A review of highway classification systems[J]. Traffic engineering & control, 1986, 27(1).
Point 6: General remark - instead of descriptions before or below figures, you can add legends - in some cases (in particular - maps) they would be more appropriate
Reply:
We have modified some Figures in the revised manuscript (Lines 260,358,451).
Round 2
Reviewer 1 Report
Thank you for incorporating the comments and making the requested corrections. I recommend accepting this manuscript for publication.